# Changes in physical activity levels, eating habits and psychological well-being during the Italian COVID-19 pandemic lockdown: Impact of socio-demographic factors on the Florentine academic population

**Gabriele Mascherini**[1]*, **Dolores Catelan**[2], **Domenico E. Pellegrini-Giampietro**[3], **Cristian Petri**[1], **Cristina Scaletti**[1], **Massimo Gulisano**[1]

1 Dipartimento di Medicina Sperimentale e Clinica, Università degli Studi di Firenze, Firenze, Italy,
2 Dipartimento di Statistica, Informatica, Applicazioni 'G. Parenti', Università degli Studi di Firenze, Firenze, Italy, 3 Dipartimento di Scienze della Salute, Università degli Studi di Firenze, Firenze, Italy

* gabriele.mascherini@unifi.it

## Abstract

The confinement and lockdown imposed by the COVID-19 pandemic have produced restrictions in the lifestyle of Italian citizens with variations in their psychological well-being. The aim of the study was to identify changes and relationship with socio-demographic parameters. An online survey was administered to 1383 subjects (1007 females and 307 males) working in the University of Florence, Italy. Three validated questionnaires were used for the survey: the Global Physical Activity Questionnaire, the Med Diet Score and the Psychological General Well-Being Index-A. All the subjects were asked to complete the questionnaires twice, in order to attain a picture of the habits before and a later time point during confinement. Our results show that work-related physical activity was decreased, along with an increase in sedentary behaviour (from 07:22±03:20 to 08:49±03:41 h:min; p<0.001, ES = 0.38), whereas recreational physical activity was increased (vigorous exercise varied from 568.5 ± 838.6 to 833.7 ± 1263.0 METs; p<0.002, ES = 0.25). Eating habits changed according to the place where meals were eaten, with an increased habit for breakfast and snacks and a slight increase in alcohol consumption. Psychological well-being decreased (Index from 21.4±3.9 to 18.0±5.3; p<0.001, ES = 0.723), especially in terms of vitality and positive thinking. The socio-demographic variables affecting these variations were mostly represented by age, gender and working conditions: young age and self-employment conditions can be considered factors for the changes in daily habits induced by confinement that may affect psychological well-being.

## Introduction

In Italy, the first two cases of COVID-19 were confirmed on January 30, 2020, while the first viral outbreak was detected by 20 February 2020. In response to these first cases, on 21

**Data Availability Statement:** All relevant data are within the paper and its Supporting Information files.

**Funding:** The authors received no specific funding for this work.

**Competing interests:** The authors have declared that no competing interests exist.

February the Italian Minister of Health issued an ordinance that decreed a mandatory quarantine for those who had been in contact with subjects tested positive for COVID-19 and active surveillance and home stay for those who had spent time in areas at risk in the previous 14 days. The further increase in positive COVID-19 cases resulted in stricter containment policies which forced all Italian citizens to be quarantined between 11 March and 3 May, 2020 [1]. During these 53 days of quarantine, common retail businesses, educational activities, and catering services were suspended, gatherings of people in public spaces were prohibited, and travel by public or private means of transport was not allowed outside of the municipalities of residency, if not for documented work or health reasons [2].

The daily habits of Italian citizens in terms of physical activity and nutrition underwent severe changes on account of the forced domestic isolation.

Physical activity levels, for example, decreased not only because of the quarantine restrictions but also because of the order of the Italian Ministry of Health, which prohibited the access to parks and playgrounds and carrying out recreational activities or outdoor games. Individual physical activities were allowed only close to home, provided that a one-meter distance was maintained between individuals. Therefore, we can expect that both the frequency and the duration of physical activity decreased, especially for those who used to regularly attend gyms or health clubs before the restrictions [3, 4].

The different organization of work schedules produced changes in eating habits, in that all meals were eaten at home rather than in restaurants, cafes or bars. It can be reasonably hypothesized that the increased time spent at home promoted a reduction in take-away meals, increased the frequency of snacks and favoured the consumption of fresh food [5].

In addition to the changes in physical activity and eating habit due to the lockdown, we also must consider the effects on psychological well-being [6, 7]. During the quarantine period in the course of the Middle Eastern respiratory syndrome (MERS) epidemic, 7% of the subjects showed symptoms of anxiety and 17% showed feelings of anger, after 4–6 months after the end of the quarantine these values were decreased to 3% and 6%, respectively [8].

The aim of the present study was to identify the changes in physical activity levels, eating habits and psychological well-being during the March-May 2020 Italian lockdown by means of an online questionnaire that all faculty, staff and students members at the University of Florence were asked to complete. Also, the socio-demographic parameters and scores possibly affecting these variations were analysed, with particular attention to the changes in psychological well-being.

## Materials and methods

### Study population

In order to study the possible changes in physical activity, eating habits and psychological conditions induced by the quarantine during the COVID-19 pandemic, a questionnaire was distributed to be completed online. All faculty, technical/administrative staff and student members of University of Florence (teachers, students and technical / administrative staff) received an invitation via email to fill in the questionnaire anonymously. Self-eligibility criteria of participation in the study were: 1) age $\geq$ 18 years-old, 2) residence in Italy and 3) access to the Internet. Self-exclusion criteria were any form of illness, participation in a strict weight-loss control program participation, pregnancy or childbirth within one year prior to the beginning of the study.

The informed consent procedure disclosed that all data would be used only for research purposes only and that the data would not be publicly accessible. All responses from attendees were anonymous and confidential according to Google's privacy policy, participants were not

required to mention their names or contact information. In addition, participants were allowed to interrupt the completion of the questionnaire at any stage without providing any type of explanation. Before completing the survey, all participants had to sign the informed consent form and agreed to declare their voluntary participation in to the anonymous study.

The study was carried out in compliance with the ethical standards of the Declaration of Helsinki of 1975 and was approved by the Ethical Commission of University of Florence (Protocol. n. 0086012, 19/06/2020).

## Procedures

All participants were first requested to provide a number of general demographic questions including age, gender, level of education, marital status, dimensions of their house, number of inhabitants, presence of open space such as a garden or terrace, loss of relatives during lockdown, occupation before and during lockdown, and self-reported height and body weight. Then, three validated questionnaires were used:

1. The Global Physical Activity Questionnaire (GPAQ) for the assessment of physical activity level,

2. The Med Diet Score for the identification of eating habits,

3. The Psychological General Well-Being Index-A (PGWBI-A) test for the evaluation of alterations in psychological states.

Each questionnaire was proposed twice, the first time to assess the habits before quarantine, the second to evaluate the changes during quarantine (S2 File).

**Global Physical Activity Questionnaire (GPAQ).** The World Health Organization developed the GPAQ for the surveillance of physical activity in various countries [9]. It includes 16 questions and collects information on physical activity participation in three different settings as well as on sedentary behaviour. Vigorous intensity activities are defined in the Questionnaire as activities that require hard physical effort and cause large increases in breathing or heart rate, whereas moderate intensity activities are those that require moderate physical effort and cause small increases in breathing or heart rate. The domains are:

- Activity at work: days per week and minutes per day of vigorous and/or moderate physical activity at work.

- Moving to and from places: days per week and minutes per day of walking or bicycle use for at least 10 min continuously to get to and from places.

- Recreational activities: with the exclusion of work and transport activities, days per week and minutes per day of vigorous and/or moderate physical activity during sports, fitness or leisure activities.

- Sedentary behaviour: time usually spent in sitting or reclining at work, at home, traveling to and from places, or with friends, including time spent reading, playing games and watching television or a personal computer. Sleep time is not included.

In order to calculate individual overall energy expenditure using the GPAQ data, 4 Metabolic Equivalents of Task (METs) were assigned to time spent in moderate activities and 8 METs to time spent in vigorous activities. Applying MET values to activity levels allowed us to calculate total physical activity. Therefore, for analysis purposes these domains were further broken down into six different "sub-domains". These "sub-domains" were vigorous work, moderate work, moving, vigorous leisure, moderate leisure, sitting [10].

**Med Diet Score.**   The Med Diet Score is a questionnaire that assesses eating habits according to the frequencies of weekly consumption of food groups [11]. The nine food groups are: non-refined cereals (e.g., whole grain bread, pasta, brown rice, etc.), fruits, vegetables, legumes, potatoes, fish, meat and meat products, poultry, full fat dairy products (e.g., cheese, yogurt, milk). Two additional food groups were represented by olive oil and alcohol. Individual ratings (from 0 to 5 or the reverse) were assigned to each food group according to their position in the Mediterranean diet pyramid. Thus, the score ranged from 0 to 55, with higher values indicating greater adherence to the Mediterranean diet.

**Psychological General Well-Being Index-A (PGWBI-A).**   The PGWBI-A can be considered as one of the first tools for the evaluation of quality of life. The final validated version, contains 22 selected items with response categories normalized according to a 6-point Likert scale, and values between 0 and 5 [12]. Six subscale scores covering the dimensions of Anxiety, Depression, Positive well-being, Self-control, General health and Vitality, and a global score (Index) can be calculated, and which providing a measure of psychological states. In 2016, in order to obtain greater acceptability, an abbreviated form reduced to six questions (one for each domain) was validated. The six questions were as follows:

1. Anxiety, "Were you generally tense or did you feel any tension during the past month?"

2. Depression, "Did you feel depressed during the past month?"

3. Positive well-being, "I felt cheerful, light-hearted during the past month."

4. Self-control, "Have you been in firm control of your behaviour, thoughts, emotions or feelings during the past month?"

5. General health, "Did you feel healthy enough to carry out the things you like to do or had to do during the past month?"

6. Vitality, "How much energy, pep, or vitality did you have or feel during the past month?"

An increasing score, ranging from 0 to 30 or higher values, indicates better psychological well-being [13].

## Statistical analysis

Descriptive statistics are reported in terms of mean and standard deviations. T-tests for paired data were used to compare before-during value of the different scores. The Cohen's d Effect Size (ES) was calculated to determine the magnitude of effect. ES was assessed using the following criteria: small < 0.20, medium < 0.50 and large < 0.80.

We analysed the during/before score differences through linear regression models. For each of the scores we generated the during/before difference. This difference is used as an outcome in the regression models. The mean of the score before and after is used to adjust for subject-specific mean score [14]. We report regression coefficients from a model with subject specific mean score. Inference based on selected samples–e.g. due to low response rate–is valid provided variables related to the selection mechanism are identified and included as confounders in the analysis [15]. The multiple regression analysis performed on the outcome variables adjusts for individual propensity, which is the most plausible determinants of the response rate.

Calculations were carried out with IBM-SPSS® version 25.0 (IBM Corp., Armonk, NY, USA, 2017) and with Stata 16 (StataCorp. 2019. Stata Statistical Software: Release 16. College Station, TX: StataCorp LLC).

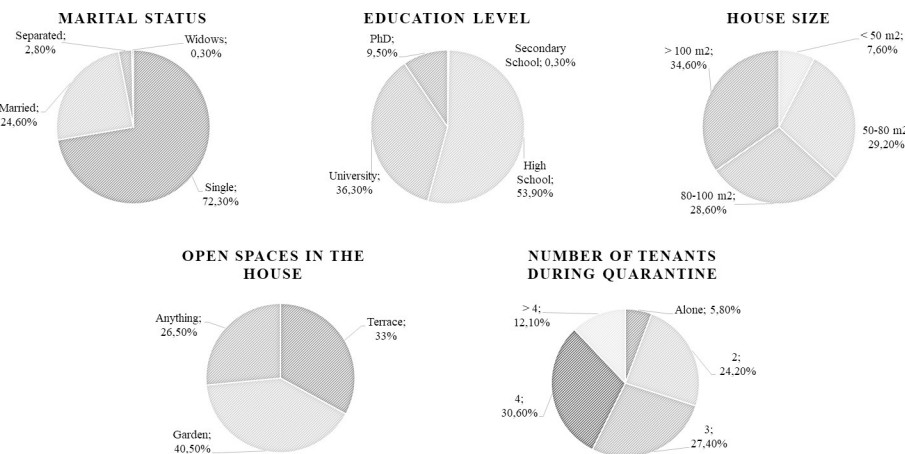

**Fig 1. This is the demographic information on the sample and housing characteristics during quarantine.**

## Results

### General characteristics of the population under study

Among the University of Florence estimated population of approximately 55000 members, 1383 subjects filled in the questionnaire (1007 females and 307 males, mean age 29.8±12.2 yrs and 33.2±14.5 yrs; respectively; 2.5% response rate). Thirty-two of them suffered the loss of a loved one due to COVID-19. Marital status, education level, house size, number of tenants during quarantine and the presence of open spaces in the house are shown in Fig 1.

During the 53 days of quarantine the body weight of the sample population increased from 64.9±13.8 to 65.3±14.1 kg (p<0.001), therefore the mean BMI increased, albeit remaining in the normal range, from 22.7 to 22.8 kg/m$^2$.

### Physical activity

Weekly physical activity levels showed a reduction in work-related and travel-related METs: prior to quarantine, 67 subjects reported vigorous physical activity at work, 220 of them reported moderate physical activities at work and 797 reported physical activities during daily commuting. During the quarantine only 26, 83 and 358 subjects, respectively, continued these types of physical activity. As for recreational activities, an increase in METs was observed under both vigorous and moderate exercise. The increase in vigorous exercise was due to an increase in the weekly frequency and duration of activities for each subject during quarantine, while the increase in moderate exercise was due to an increase in subjects who practiced this type of activity (from 692 before quarantine to 780 during quarantine). As expected, the time spent in sedentary behaviours on average increased by about 1.30 h per day (Table 1).

### Eating habits

Fig 2 shows the place where meals were consumed. The comparison between pre and post-quarantine shows an increase in domestic consumption of all meals; in particular, subjects who had breakfast, mid-morning and mid-afternoon snacks, and dinner were increased.

During the quarantine, foods such as cereals, potatoes, fruits and alcohol were selected more frequently; conversely, the choice of legumes, dairy products and red meat was decreased. However, these variations in the weekly frequency of food consumption did not alter the adherence score to the Mediterranean diet, which remained at medium-high values (Table 2).

**Table 1. This is the weekly physical activity expressed as METs at work (vigorous and moderate), during transport activities (moving) and recreational exercise (vigorous and moderate), as well as daily time spent in sedentary behaviours.** All values are compared before and during the quarantine period.

| | Before confinement | During confinement | Δ (Δ %) | 95% CI Δ | p value | Cohen's d |
|---|---|---|---|---|---|---|
| METs Vigorous work | 149.4±881.7 | 68.0±591.6 | -81.4 (-54.5) | 36.9–125.9 | <0.001 | 0.19 |
| METs Moderate work | 304.5±901.6 | 108.8±549.5 | -195.7 (-74.3) | 152.2–239.3 | <0.001 | 0.26 |
| METs Moving | 427.6±560.7 | 181.1±468.9 | -246.6 (-57.7) | 213.7–279.4 | <0.001 | 0.48 |
| METs Vigorous exercise | 568.5±838.6 | 833.7±1263.0 | 265.2 (+46.6) | 325.2–205.1 | <0.001 | 0.25 |
| METs Moderate exercise | 381.5±550.4 | 492.9±672.5 | 111.4 (+29.2) | 147.2–75.6 | <0.001 | 0.18 |
| Total METs | 1831.6±1922.7 | 1684.5±1961.9 | -147.1 (-8.1) | 39.4–254.9 | <0.01 | 0.08 |
| Sedentary behaviours (h: min/day) | 07:22±03:20 | 08:49±03:41 | 1:27 (+17.9) | 1:17–1:36 | <0.001 | 0.38 |

## Psychological well-being

The results related to psychological well-being are shown in Table 3. All the six domains that were analysed showed a reduction during the quarantine period. The main reductions were observed for the Positive well-being and to Vitality dimensions.

## Relationship with socio-demographic variables

The relationship between socio-demographic variables and levels of physical activity, eating habits and psychological well-being are shown in Table 4.

The factors that favoured an engagement in physical activity were being female, of young age, a student, being separated or divorced, or having a house with a garden. On the other hand, those factors that hindered physical activity were being a male, a self-employed or on-demand worker, being widowed, or having suffered the loss of a loved one.

The factors that promoted healthy eating habits were young age and male gender, whereas the heterogeneity of the observed results did not allow the identification of factors that led to unhealthy nutritional choices.

The main protective factors against an impairment of psychological well-being appeared to be the male gender, adulthood, a high level of education and the size of the home. On the other hand, the factors that facilitated the deterioration of psychological well-being were being of

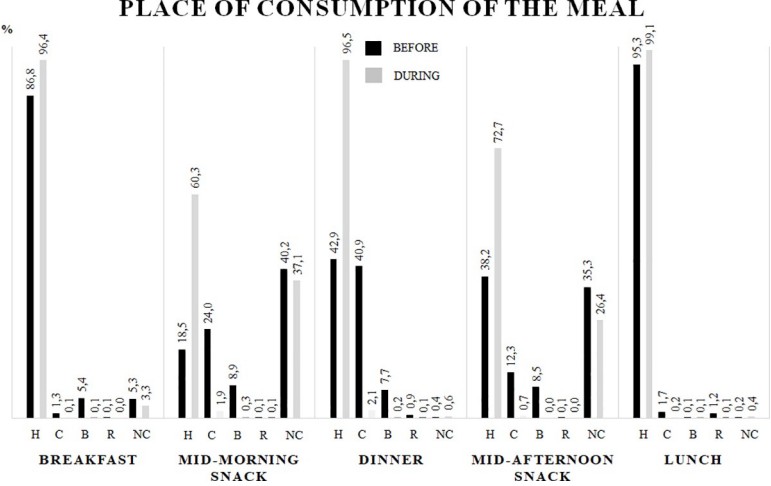

**Fig 2. These are the places where meals were consumed, as compared before and during the COVID-19 quarantine.** Data were reported as percentage of prevalence. H = Home; C = Canteen; B = Bar; R = Restaurant; NC = Not Consumed.

**Table 2. These are the frequencies of food group weekly consumption before and during the quarantine.**

|  | Before confinement | During confinement | Δ (Δ %) | 95% CI Δ | p value | Cohen's d |
|---|---|---|---|---|---|---|
| **Cereals** | 1.7±1.0 | 1.9±1.1 | 0.21 (+11.9) | 0.24–0.17 | <0.001 | 0.19 |
| **Potatoes** | 1.0±0.5 | 1.1±0.6 | 0.07 (+7.4) | 0.10–0.05 | <0.001 | 0.18 |
| **Fruits** | 2.4±1.2 | 2.5±1.2 | 0.08 (+3.4) | 0.12–0.04 | <0.001 | 0.08 |
| **Vegetables** | 2.5±1.1 | 2.5±1.1 | 0.01 (+0.4) | 0.05–0.03 | 0.283 | 0.00 |
| **Legumes** | 2.0±1.0 | 1.9±1.1 | -0.06 (-3.1) | 0.02–0.10 | 0.001 | 0.09 |
| **Fish** | 1.7±1.0 | 1.7±1.0 | 0.01 (+0.5) | 0.05–0.03 | 0.313 | 0.0 |
| **Red meat** | 3.4±1.0 | 3.5±1.0 | 0.05 (+1.3) | 0.08–0.02 | 0.002 | 0.10 |
| **White meat** | 3.7±0.8 | 3.7±0.9 | -0.05 (-1.4) | 0.02–0.08 | <0.001 | 0.0 |
| **Dairy products** | 4.1±1.1 | 4.0±1.2 | -0.17 (-4.1) | 0.13–0.21 | <0.001 | 0.09 |
| **Olive oil** | 3.7±1.6 | 3.7±1.6 | 0.0 (-0.1) | 0.03–0.04 | 0.426 | 0.0 |
| **Alcohol** | 4.8±0.7 | 4.7±0.8 | -0.07 (-1.6) | 0.03–0.11 | <0.001 | 0.13 |
| **Med Diet Score** | 31.0±4.1 | 31.1±4.3 | 0.07 (+0.2) | 0.20–0.06 | 0.153 | 0.02 |

Data are shown in a score range from 0 to 5 according to their position in the Mediterranean diet pyramid. The Med Diet Score is the sum of all items in the same column.

young age, having a large number of inhabitants at home and the occupational changes. In particular, the workers that suffered major psychological distress were those who, before the confinement, had their own business or a job that was not constant but at the request of the employer.

## Discussion

The goal of this study was to identify changes in physical activity level, eating habits and psychological well-being during the Italian COVID-19 pandemic quarantine. Faculty, administrative staff and student members of the University of Florence were invited to fill in an online questionnaire focussed on two different time point: the normal habits and behaviour before the quarantine and the changes during the limitations of the lockdown. Our results show an increase in sedentary behaviours and in recreational physical activity, moderate changes in eating habits, and a deterioration in psychological well-being.

An increase in obesity was hypothesized following the pandemic, with a consequent increase in cardiovascular risk. For preventive purposes, the literature had therefore promoted healthy eating and physical activity at home during the quarantine period [16, 17]. However, the results of the present study demonstrate a mean weight gain after the quarantine period of only 0.4±2.8 kg in 53 days. It should be considered that our sample population presented initial

**Table 3. This is the psychological general well-being index before and during the COVID-19 quarantine period.**

|  | Before confinement | During confinement | Δ (Δ %) | 95% CI Δ | p value | Cohen's d |
|---|---|---|---|---|---|---|
| **Anxiety** | 2.6±1.2 | 2.3±1.4 | -0.34 (-12.8) | 0.3–0.4 | <0.001 | 0.23 |
| **Depression** | 4.3±0.8 | 3.9±1.1 | -0.44 (-10.1) | -0.4–0.5 | <0.001 | 0.41 |
| **Positive well-being** | 3.1±0.9 | 2.3±0.9 | -0.77 (-25.2) | -0.7–0.8 | <0.001 | 0.83 |
| **Self-control** | 3.6±1.1 | 2.9±1.3 | -0.73 (-20.2) | 0.6–0.8 | <0.001 | 0.56 |
| **General health** | 4.1±0.8 | 3.7±1.1 | -0.39 (-9.5) | 0.3–0.5 | <0.001 | 0.41 |
| **Vitality** | 3.7±0.8 | 2.9±1.2 | -0.81 (-21.8) | 0.7–0.9 | <0.001 | 0.79 |
| **Index** | 21.4±3.9 | 18.0±5.3 | -3.47 (-16.2) | 3.2–3.8 | <0.001 | 0.73 |

* = p value < 0.001.

**Table 4. These are the results related to the relationship between physical activity, eating habits and psychological well-being with socio-demographic variables.**

| | | Physical Activity | | Eating Habits | | Psychological well-being | |
|---|---|---|---|---|---|---|---|
| | | Coefficient | CI 95% | Coefficient | CI 95% | Coefficient | CI 95% |
| **Gender** | Male vs Female | -313.1 | -555.6;-70.5 | 0.16 | -0.14;0.46 | 0.56 | -0.06;1.18 |
| **Age (linear)** | | | | -0.01 | -0.02;0.001 | 0.40 | 0.02;0.06 |
| **Age (> median)** | Young vs Old | 220.2 | 4.5;435.9 | 0.38 | 0.11;0.65 | -1.19 | -1.74;-0.65 |
| **Marital status** | Single | Ref | | Ref | | Ref | |
| | Married | -57.5 | -309.7;194.6 | -0.28 | -0.59;0.03 | 1.03 | 0.39;1.68 |
| | Separated | 497.3 | -167.4;1161.9 | 0.43 | -0.39;1.25 | 1.55 | -0.12;3.22 |
| | Widow | -1012.7 | -3022.6;997.3 | -1.21 | -3.69;1.26 | -0.36 | -5.43;4.71 |
| **Education level** | High school | Ref | | Ref | | Ref | |
| | Graduate | -218.5 | -450.4;13.3 | -0.02 | -0.31;0.27 | 0.43 | -0.16;1.01 |
| | PhD | -127.4 | -508.4;253.6 | 0.34 | -0.14;0.81 | 1.57 | 0.61;2.54 |
| **Inhabitants in house (linear)** | | 51.7 | -45.8;149.2 | -0.01 | -0.14;0.11 | -0.26 | -0.51;-0.02 |
| **Dimension of the house (linear)** | | 102.6 | -9.2;214.3 | 0.16 | 0.03;0.30 | 0.22 | -0.06;0.50 |
| **Open space in house** | Nothing | Ref | | Ref | | Ref | |
| | Terrace | 111.7 | -169.5;393.0 | 0.01 | -0.34;0.36 | 0.11 | -0.53;0.75 |
| | Garden | 262.4 | -6.9;531.7 | -0.01 | -0.34;0.32 | -0.51 | -1.23;0.20 |
| **Job during lockdown** | Smart working | Ref | | Ref | | Ref | |
| | Students | 266.7 | 24.4;509.1 | -0.03 | -0.34;0.27 | -0.73 | -1.35;-0.11 |
| | Others | -402.7 | -696.8;-108.6 | -0.02 | -0.38;0.34 | -0.93 | -1.67;-0.19 |
| **Loss of a loved one** | No vs Yes | -1173.9 | -1891.1;-456.8 | -0.61 | -1.49;0.27 | 0.67 | -1.14;2.50 |
| **Job before lockdown** | Permanent employee | Ref | | Ref | | Ref | |
| | Fixed-term employee | -276.0 | -691.8;139.8 | 0.23 | -0.29;0.74 | -0.45 | -1.50;0.60 |
| | Self employed | -1018.3 | -1543.5;-493.1 | -0.03 | -0.68;0.62 | -1.23 | -2.56;0.10 |
| | Worker on call | -611.0 | -1205.7;-16.3 | 0.51 | -0.23;1.25 | -2.59 | -4.11;-1.08 |
| | Stage | 136.8 | -740.5;1014.0 | 0.06 | -1.03;1.15 | -1.08 | -3.31;1.14 |
| | Student | 110.2 | -153.0;373.4 | 0.14 | -0.19;0.47 | -1.27 | -1.94;-0.60 |
| | Unemployed | -552.2 | -1603.9;499.7 | 0.28 | -1.02;1.59 | 0.52 | -2.16;3.20 |
| | Retired | -163.5 | -1509.4;1182.4 | 0.23 | -1.45;1.9 | -2.26 | -5.68;1.15 |

Regression coefficients and 95% Confidence Intervals (CI) of linear regression analysis on physical activity, eating habits and psychological well-being score. All analysis are adjusted for subject-specific mean psychological score (mean of during-before score).

conditions of normal weight, characterized by both eating habits that meet the criteria of the Mediterranean diet, and relatively high levels of physical activity.

## Physical activity

As expected, during the quarantine we observed reductions in physical activity related to work duties and daily transfers, with a concomitant increase in sedentary lifestyle. Because of the extra free time from work commitments, we also observed a parallel increase in recreational physical activity, in confirmation of Di Renzo et al. [18]. Specifically, vigorous physical activity was increased both in subjects who were already carrying it out before the quarantine and in subjects who decided to start a program of moderate physical activity during the quarantine. These results partially confirm a recent study on the psychological mechanisms underlying the practice of physical activity during the Italian lockdown [7]. The intention to carry out physical activity can be considered both a self-determined process, as demonstrated by the increase in levels of vigorous activity in those who practiced it even before the lockdown, and deliberative, as demonstrated by the increase in moderate activity levels in those who did not engage in any

physical activity prior to the block [7]. In addition, we analysed the pre- and post-quarantine sedentary behaviours of the subjects. We can define a subject as sedentary if the number of hours of inactivity per day is > 7 [19]. The risk to be classified as sedentary during lockdown was demonstrated by this study to be around 5 (raw Odds Ratio OR = 4.68, CI 95%: 3.52; 6.31).

## Eating habits

Domestic confinement also changed the eating habits. The prevalence of eating meals at home was definitely increased, with a larger number of subjects having breakfast and the two mid-morning and mid-afternoon snacks at home. Our results confirms previous data on the adherence to the Mediterranean diet of the Italian population during the COVID-19 lockdown [18]. The Mediterranean diet score did not change globally, but there was an improvement in the choice of cereals, potatoes and fruits that was associated with a slight increase in the consumption of alcohol.

The variations that we observed were rather moderate, which appears to be a common feature of studies on population lifestyle. In fact, the large standard deviations of our results describe a high variability in the habits of the subjects under study.

## Psychological well-being

The COVID-pandemic by itself can produce acute panic, anxiety, obsessive behaviours, hoarding, paranoia and depression, disorders than can be exacerbated by the forced quarantine [20]. The PGWBI-A measures the self-representation of well-being and the emotional and affective distress, is sensitive to changes and is particularly useful with repeated measurements [13]. The results obtained in the present study reveal a worsening of all of the six domains investigated (Anxiety, Depression, Positive well-being, Self-control, General Health and Vitality) which results in an overall reduction of the final Index. The Index before and during the domestic confinement decreased significantly by 15.1%, an effect that was greater than that observed in other multicentre studies of similar samplings reporting a reduction in psychological well-being of 9.4% [21]. It should be noted that the dimensions of Positive well-being and Vitality displayed the largest reductions. Hence, it appears from our study that positive thoughts and perceived energy were significantly affected during the 53 days of home confinement.

## Relationship with socio-demographic variables

A recent study on Italian college students has shown that physical exercise promotes healthy food choices and that by adopting both of these measures during quarantine it is possible to obtain a positive effect on mood states [22]. Although the relationship between physical exercise, nutrition and psychological well-being have been already investigated [23], the aim of our study was also to correlate the changes in physical activity, eating habits and psychological distress during the COVID-19 quarantine with a number of socio-demographic variables. Interestingly, a recent study conducted on a larger sample of our same population demonstrated the protective effect of physical activity against depression during the COVID-19 quarantine [24]. The present study revealed modest changes in the eating habits of interviewed subjects, an overall reduction in their physical activity levels but, at the same time, an increase in recreational physical activity. Therefore, we examined the relationship between socio-demographic variables and the levels of physical activity, detecting that a job occupation acts like a barrier whereas the student status allows greater levels of exercise.

The analysis of the relationship between socio-demographic variables and psychological well-being during confinement showed that adult males experienced minor psychological suffering as compared to females. In this context, marriage played a detrimental role since separated subjects reported to be affected by psychological suffering to a lesser extent. The examination of the living conditions showed that home dimensions had greater significance than the possibility of having outdoor spaces, probably because they allowed to maintain one's privacy regardless of the number of inhabitants in each house. As for the variations induced by working conditions, self-employed has to be considered at higher risk of psychological distress. Finally, we must consider that the suffering manifested by the students is at least in part ascribable also to their young age.

A major strength of this study is the use of three validated questionnaires for each research area in order to obtain reproducible data. However, we must also consider some limitations of the study: firstly, a selection bias, that is the heterogeneity of the sample due to its relatively small size and the different University roles of the participants; secondly, a gender discrepancy in the study sample; and finally, the likelihood of biased negative thoughts when asked to remember happy and serene moments in life.

Due to the restrictions of the COVID-19 pandemic, our questionnaires were administered online; therefore, the possible approximation of to this type of administration needs to be considered and corrected in future studies. Future studies could also target specific categories of members of the universities or other companies. The sample size seems to partially compensate for the specificity of sample as far as the job and the residency site are concerned; however, our results and conclusions cannot be generalized to the general population belonging of other territories.

## Conclusions

In accord with the likelihood of a lifestyle deterioration during quarantine, the sample examined showed changes in physical activity and eating habits. The expected increase in sedentary lifestyles was in some cases compensated by the use of a larger amount of free time to exercise or by healthier food choices. However, psychological well-being appears to be impaired mainly in young people and when working conditions change. Particularly self-employment conditions are more likely to expose people to a deterioration of psychological well-being.

## Supporting information

**S1 File. Results of questionnaire.**
(XLSX)

**S2 File. ABITUD-19 questionnaire.**
(DOCX)

## Author Contributions

**Conceptualization:** Gabriele Mascherini, Domenico E. Pellegrini-Giampietro.

**Data curation:** Gabriele Mascherini, Dolores Catelan.

**Formal analysis:** Dolores Catelan.

**Investigation:** Gabriele Mascherini, Domenico E. Pellegrini-Giampietro, Cristina Scaletti.

**Methodology:** Gabriele Mascherini, Domenico E. Pellegrini-Giampietro, Cristian Petri.

**Project administration:** Gabriele Mascherini, Massimo Gulisano.

**Supervision:** Massimo Gulisano.

**Validation:** Cristian Petri.

**Visualization:** Domenico E. Pellegrini-Giampietro, Cristian Petri, Cristina Scaletti, Massimo Gulisano.

**Writing – original draft:** Gabriele Mascherini, Dolores Catelan.

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
