## [Decision Letter · Decision Letter 0]

12 Apr 2021

PONE-D-21-10183

Changes in psychological well-being, physical activity levels and eating habits during the Italian lockdown: influence of socio-demographic factors on the Florentine academic population.

PLOS ONE

Dear Dr. Mascherini,

Thank you for submitting your manuscript to PLOS ONE. After careful consideration, we feel that it has merit but does not fully meet PLOS ONE’s publication criteria as it currently stands. Therefore, we invite you to submit a revised version of the manuscript that addresses the points raised during the review process.

We look forward to receiving your revised manuscript.

Kind regards,

Cristina Cortis, Ph.D.

Academic Editor

PLOS ONE

Additional Editor Comments:

As you see from the reviewers' comments there are few suggestions that should be addressed before the paper can be finally accepted for publication. Make sure you carefully check the english writing as the journal does not provide language editing. 

Journal Requirements:

4. Please upload an English language copy of the questionnaire as a supplemental file.

Reviewers' comments:

Reviewer's Responses to Questions

**Comments to the Author**

1. Is the manuscript technically sound, and do the data support the conclusions?

Reviewer #1: Yes

Reviewer #2: Yes

2. Has the statistical analysis been performed appropriately and rigorously? 

Reviewer #1: Yes

Reviewer #2: Yes

3. Have the authors made all data underlying the findings in their manuscript fully available?

Reviewer #1: Yes

Reviewer #2: Yes

4. Is the manuscript presented in an intelligible fashion and written in standard English?

Reviewer #1: No

Reviewer #2: Yes

5. Review Comments to the Author

Reviewer #1: The manuscript topic is valuable however the English could be improved before it can be accepted for publication.

Line 51: The further increase in positive COVID-19 cases, resulted in stricter containment policies which forced all Italian citizens to be quarantined between 11 March-to 3 May, 2020.

Line 54: no new paragraph

Line 58: new paragraph:

Line 66: ?The different organization of work….that statement is confusing, consider revising.

Line 67. Consider revising. It can hypothesized that the increased time spent at home resulted in a reduction of takeaway meals and increased the frequency of snack and consumption of fresh food.

Line 70; witch – which. Another aspect witch is to considered, being related to eating habits and physical activity, is the effect on psychological well-being.

Consider; In addition to changes in eating habit and physical activity due to the lockdown, is the effect on psychological well-being.

Line:75-77. Consider: ……through an online questionnaire that all students and personnel at the University of Florence were asked to complete.

Line 90-96. Consider revising and simplify.

Line 101: Following general demographic questions (age, gender, level of

education, marital status, dimension of the house, presence of open space on the house, loss of

relatives during lockdown, number of inhabitants in the house job before and during lockdown),

three validated questionnaires were used:

Line 161:L T-tests

Line 166: generated

Line 231: not sure what ‘separation’ means.

DISCUSSION SECTION: no line numbers.

The results of the present study show a weight gain after the quarantine period of about 0.5 kg in 53 days.

(where were the body weights recorded and by whom?) This was not mentioned in the methods section..

Reviewer #2: Dear Prof. Dr. Cristina Cortis,

thank you very much for the possibility to serve as a reviewer in a prestigious periodical like Plos One.

About this paper titled “Changes in psychological wellbeing physical activity levels and eating habits during the Italian lockdown influence of sociodemographic factors on the Florentine academic population”, the contents and the rhetoric by which it was handled are appreciable and the paper is very well organized. Moreover, the analyzed topic is original and current.

Overview:

The objective of this study is to explore changes in psychological well-being, physical activity levels and eating habits during the Italian first lockdown in people belonging to the University of Florence. 1383 between teachers, students and technical / administrative staff filled out the questionnaire answering about marital status, education level, house size, number of tenants during quarantine, the presence of open spaces in the house, psychological well-being, physical activity levels and eating habits before and during the “stay-at-home” measures.

Data collected indicate that during the restrictions there was a general lifestyle deterioration. Indeed, sedentary lifestyles increased and phycological well-being decreased even though exercise in free time and the consume of healthier food increased.

General comment:

Nevertheless, this paper examining a trend topic and the study is original, some minor questions require clarification in order to improve the quality of the manuscript. I have listed below specific comments to the authors.

Material and Methods:

Lines 87 and 102: Participants had to answer about residence and house characteristics. Was taken into account also where the house was located (e.g. city, hill, mountains)? Should be a discriminating factor for phycological well-being?

Procedures:

Line 95 “to voluntary”: I suggest to remove “to”.

Line 103. “Number of inhabitants in the house” and Line 106. “eating habits”. There was a question (for the students) about living with parents? It’s possible for the students who lived with parents had different eating habits compared with who lived alone or with other students? It’s possible a sub analysis?

Lines 108 – 109 “each questionnaire was proposed twice, the first time in relation to habits before quarantine ..”. I think this design can generate a bias. Indeed, if I think, about habits before a worst period probably my habits appear better in every field. This doesn’t discredit the findings but I suggest to add a paragraph in limitations session.

Results:

Line 176 “2.5% response rate”: I suggest to add a possible rationale about this data. 1383 questionnaire collected are a great number, but the response rate is very low.

Table 2: In this table are presented the frequencies of weekly consumption of food groups and the differences between before and during restrictions. Did the authors collect data about calories? I think that frequencies are important but also quantities.

6. PLOS authors have the option to publish the peer review history of their article (what does this mean?). If published, this will include your full peer review and any attached files.

Reviewer #1: No

Reviewer #2: No

---

## [Author Response · Author response to Decision Letter 0]

23 Apr 2021

Additional Editor Comments:

As you see from the reviewers' comments there are few suggestions that should be addressed before the paper can be finally accepted for publication. Make sure you carefully check the english writing as the journal does not provide language editing. 

Dear editor, thank you for the work you have done.

The response letter to the reviewers was made after each point-by-point observation. Responses are highlighted in red.

A native English speaker proofread the revised version of the manuscript.

Journal Requirements:

The revised version has made changes in accordance with the recommendations of the journal.

The authors revised the reference list by updating:

- reference 3: the URL and doi that were not yet available for the above prepress have been added

- reference 16: the published version has been inserted instead of ahead print version

Captions for your Supporting Information files were included at the end of the manuscript, which coincide with those within the text.

4. Please upload an English language copy of the questionnaire as a supplemental file.

The English version of the questionnaire has been uploaded as supplemental file (S2 File)

Reviewers' comments:

Reviewer's Responses to Questions

Comments to the Author

1. Is the manuscript technically sound, and do the data support the conclusions?

Reviewer #1: Yes

Reviewer #2: Yes

2. Has the statistical analysis been performed appropriately and rigorously?

Reviewer #1: Yes

Reviewer #2: Yes

3. Have the authors made all data underlying the findings in their manuscript fully available?

Reviewer #1: Yes

Reviewer #2: Yes

4. Is the manuscript presented in an intelligible fashion and written in standard English?

Reviewer #1: No

Reviewer #2: Yes

A native English speaker proofread the revised version of the manuscript.

5. Review Comments to the Author

Reviewer #1: The manuscript topic is valuable however the English could be improved before it can be accepted for publication.

The authors thank the reviewers for the work done.

Below are the answers point by point to the observations made in red.

In addition, the requested changes have been made in red in the text. 

A native English speaker proofread the revised version of the manuscript.

Line 51: The further increase in positive COVID-19 cases, resulted in stricter containment policies which forced all Italian citizens to be quarantined between 11 March-to 3 May, 2020.

The text has been changed accordingly

Line 54: no new paragraph

The new paragraph has been removed

Line 58: new paragraph:

A new paragraph has been added

Line 66: ?The different organization of work….that statement is confusing, consider revising.

A native English speaker proofread the revised version of the manuscript.

Line 67. Consider revising. It can hypothesized that the increased time spent at home resulted in a reduction of takeaway meals and increased the frequency of snack and consumption of fresh food.

Yes, the reviewer interpreted correctly. This hypothesis derives from a previous published study conducted on the Spanish population doi: 10.3390/nu12092826.

However, a native English speaker proofread the revised version of the manuscript.

Line 70; witch – which. Another aspect witch is to considered, being related to eating habits and physical activity, is the effect on psychological well-being.

Consider; In addition to changes in eating habit and physical activity due to the lockdown, is the effect on psychological well-being.

Thanks, I apologize for the typo. The sentence was replaced with the one suggested by the reviewer

Line:75-77. Consider: ……through an online questionnaire that all students and personnel at the University of Florence were asked to complete.

The sentence was replaced with the one suggested by the reviewer

Line 90-96. Consider revising and simplify.

A native English speaker proofread the revised version of the manuscript.

Line 101: Following general demographic questions (age, gender, level of

education, marital status, dimension of the house, presence of open space on the house, loss of

relatives during lockdown, number of inhabitants in the house job before and during lockdown),

three validated questionnaires were used:

The sentence was replaced with the one suggested by the reviewer and revised by expert

Line 161:L T-tests

Modified

Line 166: generated

Modified

Line 231: not sure what ‘separation’ means.

The authors intended "to be separated or to be divorced". Therefore, the term was replaced with “being separated or divorced”

DISCUSSION SECTION: no line numbers.

The results of the present study show a weight gain after the quarantine period of about 0.5 kg in 53 days.

(where were the body weights recorded and by whom?) This was not mentioned in the methods section..

A sentence in Procedures section has been added “…..and self-reported height and body weight”

Reviewer #2: Dear Prof. Dr. Cristina Cortis,

thank you very much for the possibility to serve as a reviewer in a prestigious periodical like Plos One.

About this paper titled “Changes in psychological wellbeing physical activity levels and eating habits during the Italian lockdown influence of sociodemographic factors on the Florentine academic population”, the contents and the rhetoric by which it was handled are appreciable and the paper is very well organized. Moreover, the analyzed topic is original and current.

The authors thank the reviewers for the work done. Below are the answers point by point to the observations made in red. In addition, the requested changes have been made in red in the text. A native English speaker proofread the revised version of the manuscript.

Overview:

The objective of this study is to explore changes in psychological well-being, physical activity levels and eating habits during the Italian first lockdown in people belonging to the University of Florence. 1383 between teachers, students and technical / administrative staff filled out the questionnaire answering about marital status, education level, house size, number of tenants during quarantine, the presence of open spaces in the house, psychological well-being, physical activity levels and eating habits before and during the “stay-at-home” measures.

Data collected indicate that during the restrictions there was a general lifestyle deterioration. Indeed, sedentary lifestyles increased and phycological well-being decreased even though exercise in free time and the consume of healthier food increased.

General comment:

Nevertheless, this paper examining a trend topic and the study is original, some minor questions require clarification in order to improve the quality of the manuscript. I have listed below specific comments to the authors.

Material and Methods:

Lines 87 and 102: Participants had to answer about residence and house characteristics. Was taken into account also where the house was located (e.g. city, hill, mountains)? Should be a discriminating factor for phycological well-being?

The authors thank the reviewer for the comment. This aspect was not included in the questionnaire questions. By subjecting the sample to 3 questionnaires referring to 2 times (therefore 6 questionnaires) the general questions were reduced to a minimum in order not to burden the survey. In detail, the location of the house was not included as physical activity was only allowed near the house during the lockdown.

Procedures:

Line 95 “to voluntary”: I suggest to remove “to”.

Modified

Line 103. “Number of inhabitants in the house” and Line 106. “eating habits”. There was a question (for the students) about living with parents? It’s possible for the students who lived with parents had different eating habits compared with who lived alone or with other students? It’s possible a sub analysis?

The authors thank the reviewer for comment and for suggesting further analysis. However, there were no specific questions addressed to off-site students. It would probably have been appropriate to propose a specific questionnaire to the students, but this would have reduced the sample size. This could be useful for future study directions. However, a sentence has been added at the end of discussion section “Future studies could also target specific categories of members of the universities or other companies”

Lines 108 – 109 “each questionnaire was proposed twice, the first time in relation to habits before quarantine ..”. I think this design can generate a bias. Indeed, if I think, about habits before a worst period probably my habits appear better in every field. This doesn’t discredit the findings but I suggest to add a paragraph in limitations session.

The authors thank the reviewer for the comment. A sentence has been added in discussion section. “and finally, the likelihood of biased negative thoughts when asked to remember happy and serene moments in life.”

Results:

Line 176 “2.5% response rate”: I suggest to add a possible rationale about this data. 1383 questionnaire collected are a great number, but the response rate is very low.

The authors agree with the reviewer. However, it should be considered that during the first part of the pandemic, where containment measures were particularly high, research activities through online questionnaires increased. This has led to numerous invitations to fill in online surveys being sent: the response rate may be further reduced due to the high demand for participation in multiple surveys.

However, recent studies report how carrying out a selection process through cohort studies allows good reproducibility of the results regardless of the response rate. In support of this thesis, the authors report a sentence taken from the paper Richiardi L, Pearce N, Pagano E, Di Cuonzo D, Zugna D, Pizzi C. Baseline selection on a collider: a ubiquitous mechanism occurring in both representative and selected cohort studies. J Epidemiol Community Health. 2019 May;73(5):475-480. doi: 10.1136/jech-2018-211829:

 “We conclude that, when conducting a cohort study, different source populations, whether ’selected’ or ’representative’, may lead to different exposure–outcome risk factor associations, and thus different degrees of lack of exchangeability, but that one approach is not inherently more or less biased than the other. The key issue is whether the relevant risk factors can be identified and controlled.”

In view of this, the authors may believe that they have identified and controlled the variables studied using three validated questionnaires.

Table 2: In this table are presented the frequencies of weekly consumption of food groups and the differences between before and during restrictions. Did the authors collect data about calories? I think that frequencies are important but also quantities.

The authors thank the reviewer for the comment. The purpose of the study refers to the quality of eating habits and adherence to the Mediterranean diet rather than quantity. Furthermore, the assessment of calorie intake through a self-reported online questionnaire would have required numerous questions (e.g. EatWellQ8 146-item FFQ), accompanied by images of the portions consumed. This would have burdened the compilation of the questionnaire, also in view of the double compilation (pre-covid and quarantine). Ultimately, this choice would have provided a low to moderate agreement for the measurement of energy intake.

---

## [Decision Letter · Decision Letter 1]

4 May 2021

PONE-D-21-10183R1

Changes in physical activity levels, eating habits and psychological well-being during the Italian COVID-19 pandemic lockdown: impact of socio-demographic factors on the Florentine academic population.

PLOS ONE

Dear Dr. Mascherini,

Thank you for submitting your manuscript to PLOS ONE. After careful consideration, we feel that it has merit but does not fully meet PLOS ONE’s publication criteria as it currently stands. Therefore, we invite you to submit a revised version of the manuscript that addresses the points raised during the review process.

As you can see from the comments, the paper is almost ready to be published. Before resubmitting the manuscript, please carefully check it for syntax and grammar, especially because PlosOne does not copyedit papers, as suggested by Reviewer 1.

We look forward to receiving your revised manuscript.

Kind regards,

Cristina Cortis, Ph.D.

Academic Editor

PLOS ONE

Journal Requirements:

Reviewers' comments:

Reviewer's Responses to Questions

**Comments to the Author**

1. If the authors have adequately addressed your comments raised in a previous round of review and you feel that this manuscript is now acceptable for publication, you may indicate that here to bypass the “Comments to the Author” section, enter your conflict of interest statement in the “Confidential to Editor” section, and submit your "Accept" recommendation.

Reviewer #1: All comments have been addressed

Reviewer #2: (No Response)

2. Is the manuscript technically sound, and do the data support the conclusions?

Reviewer #1: Yes

Reviewer #2: Yes

3. Has the statistical analysis been performed appropriately and rigorously? 

Reviewer #1: Yes

Reviewer #2: Yes

4. Have the authors made all data underlying the findings in their manuscript fully available?

Reviewer #1: Yes

Reviewer #2: Yes

5. Is the manuscript presented in an intelligible fashion and written in standard English?

Reviewer #1: Yes

Reviewer #2: Yes

6. Review Comments to the Author

Reviewer #1: There are a few minor language issues but the revised manuscript is much stronger and ready for publication.

Reviewer #2: Dear Prof. Dr. Cristina Cortis,

thank you very much for the possibility to re-serve as a reviewer in a prestigious periodical like Plos One.

Also, I would like to commend the authors for their revision work. There is only one minor request, that I would like to reiterate. I think that adds value to the study.

The authors answered my question about response rate by emphasizing the use of three validated questionaries. This certainly shows the validity of the work, but I believe that what the authors reported in their reply to my comment should also be reported in the text at Line 176. I re-suggest to add a possible rationale about this data.

7. PLOS authors have the option to publish the peer review history of their article (what does this mean?). If published, this will include your full peer review and any attached files.

Reviewer #1: No

Reviewer #2: No

---

## [Author Response · Author response to Decision Letter 1]

5 May 2021

Review Comments to the Author

Dear Editor and Reviewers, thank you for the work done in order to improve our manuscript.

The response letter to the reviewers was made after each point-by-point observation. Responses are highlighted in red.

Reviewer #1: There are a few minor language issues but the revised manuscript is much stronger and ready for publication.

Line 29: the aim of the study was to identify changes in psychological well being and relationship with socio-demographic parameters. 

Thank you for the proposal, corrections have been made accordingly.

Line 36: change were to was 

Thank you for the proposal, corrections have been made accordingly.

Line 44-45: can be considered factors for the changes……(delete the word risk)

Thank you for the proposal, corrections have been made accordingly.

Line 148-149 (not sure if the words Metabolic Equivalents need to be capitalized)

In order to provide the reader with an overall definition of the abbreviation, the authors prefer to leave it in its entirety by including also the term TASK that completes the abbreviation MET.

Line 225: Figure 2 shows the place where meals were consumed.

Thank you for the proposal, corrections have been made accordingly.

Line 226: The comparison between pre and post-quarantine shows………..

Thank you for the proposal, corrections have been made accordingly.

Discussion Section:

This needs to be re-written for clarity.

Because an increase in obesity has been was hypothesized following the pandemic, with a consequent increase in cardiovascular risk, the scientific For preventive purposes, the literature was had therefore geared towards promoted preventive measures such as healthy eating and physical activity at home during the quarantine period [15, 16]. 

Thank you for the proposal, the actual sentence is: “An increase in obesity was hypothesized following the pandemic, with a consequent increase in cardiovascular risk. For preventive purposes, the literature had therefore promoted healthy eating and physical activity at home during the quarantine period.”

However, psychological well-being appears to be impaired mainly in people of affected by young age and when working conditions change.

Consider changing to 

However, psychological well-being appears to be impaired mainly in young people of when working conditions change.

Thank you for the proposal, the actual sentence is: “However, psychological well-being appears to be impaired mainly in young people and when working conditions change.”

Reviewer #2: Dear Prof. Dr. Cristina Cortis,

thank you very much for the possibility to re-serve as a reviewer in a prestigious periodical like Plos One.

Also, I would like to commend the authors for their revision work. There is only one minor request, that I would like to reiterate. I think that adds value to the study.

The authors answered my question about response rate by emphasizing the use of three validated questionaries. This certainly shows the validity of the work, but I believe that what the authors reported in their reply to my comment should also be reported in the text at Line 176. I re-suggest to add a possible rationale about this data.

Thank you for the proposal. As suggest by reviewer, a sentence has been added at line 173, just before the version of the software used for statistical analysis:

“Inference based on selected samples – e.g. due to low response rate – is valid provided variables related to the selection mechanism are identified and included as confounders in the analysis [15]. The multiple regression analysis performed on the outcome variables adjusts for individual propensity, which is the most plausible determinants of the response rate.”

---

## [Decision Letter · Decision Letter 2]

17 May 2021

Changes in physical activity levels, eating habits and psychological well-being during the Italian COVID-19 pandemic lockdown: impact of socio-demographic factors on the Florentine academic population.

PONE-D-21-10183R2

Dear Dr. Mascherini,

We’re pleased to inform you that your manuscript has been judged scientifically suitable for publication and will be formally accepted for publication once it meets all outstanding technical requirements.

Kind regards,

Cristina Cortis, Ph.D.

Academic Editor

PLOS ONE

Additional Editor Comments (optional):

The authors successfully dealt with the required revisions and addressed all comments raised by the Reviewers.

Therefore, I think the paper could be now accepted for publication. 

Reviewers' comments:

Reviewer's Responses to Questions

**Comments to the Author**

1. If the authors have adequately addressed your comments raised in a previous round of review and you feel that this manuscript is now acceptable for publication, you may indicate that here to bypass the “Comments to the Author” section, enter your conflict of interest statement in the “Confidential to Editor” section, and submit your "Accept" recommendation.

Reviewer #1: All comments have been addressed

Reviewer #2: All comments have been addressed

2. Is the manuscript technically sound, and do the data support the conclusions?

Reviewer #1: Yes

Reviewer #2: Yes

3. Has the statistical analysis been performed appropriately and rigorously? 

Reviewer #1: Yes

Reviewer #2: Yes

4. Have the authors made all data underlying the findings in their manuscript fully available?

Reviewer #1: Yes

Reviewer #2: Yes

5. Is the manuscript presented in an intelligible fashion and written in standard English?

Reviewer #1: Yes

Reviewer #2: Yes

6. Review Comments to the Author

Reviewer #1: This re-write reads considerably better than the original submission. Excellent work by the authors to improve readability.

Reviewer #2: Dear Editor and Authors,

I appreciate the author’s work, and reply. All answers satisfy my requests.

7. PLOS authors have the option to publish the peer review history of their article (what does this mean?). If published, this will include your full peer review and any attached files.

Reviewer #1: No

Reviewer #2: No

---

## [Editor Report · Acceptance letter]

19 May 2021

PONE-D-21-10183R2 

Changes in physical activity levels, eating habits and psychological well-being during the Italian COVID-19 pandemic lockdown: impact of socio-demographic factors on the Florentine academic population. 

Dear Dr. Mascherini:

I'm pleased to inform you that your manuscript has been deemed suitable for publication in PLOS ONE. Congratulations! Your manuscript is now with our production department. 

Kind regards, 

on behalf of

Prof. Dr. Cristina Cortis 

Academic Editor

PLOS ONE